# Finding multiple reaction pathways via global optimization of action

Juyong Lee[1], In-Ho Lee[2,3], InSuk Joung[3,4], Jooyoung Lee[3,4] & Bernard R. Brooks[1]

Global searching for reaction pathways is a long-standing challenge in computational chemistry and biology. Most existing approaches perform only local searches due to computational complexity. Here we present a computational approach, Action-CSA, to find multiple diverse reaction pathways connecting fixed initial and final states through global optimization of the Onsager–Machlup action using the conformational space annealing (CSA) method. Action-CSA successfully overcomes large energy barriers via crossovers and mutations of pathways and finds all possible pathways of small systems without initial guesses on pathways. The rank order and the transition time distribution of multiple pathways are in good agreement with those of long Langevin dynamics simulations. The lowest action folding pathway of FSD-1 is consistent with recent experiments. The results show that Action-CSA is an efficient and robust computational approach to study the multiple pathways of complex reactions and large-scale conformational changes.

[1] Laboratory of Computational Biology, National Heart, Lung, and Blood Institute (NHLBI), National Institutes of Health (NIH), Bethesda, Maryland 20892, USA. [2] Center for Materials Genome, Korea Research Institute of Standards and Science, Daejeon 34113, Republic of Korea. [3] Center for In Silico Protein Science, School of Computational Science, Korea Institute for Advanced Study, Seoul 02455, Republic of Korea. [4] School of Computational Sciences, Korea Institute for Advanced Study, Seoul 02455, Republic of Korea. Correspondence and requests for materials should be addressed to Juyong L. (email: juyong.lee@nih.gov) or to Jooyoung L. (email: jlee@kias.re.kr) or to B.R.B. (email: brb@nih.gov).

Finding multiple plausible reaction pathways between two end states is a long-standing challenge in computational sciences[1]. One of the common approaches is to perform long-time molecular dynamics (MD) simulations. Despite recent advances in the MD methodologies and computational technologies, this approach suffers from a timescale problem. Many biological reactions such as protein folding and protein conformational transitions occur in the microsecond or millisecond ranges, which are still hard to be performed even with the fastest computers available today. Also, MD simulations starting from one end state are not guaranteed to reach the other end state of interest especially considering the inaccuracies of current force fields. Thus, developing an efficient computational method to find multiple possible reaction pathways connecting two end states can serve as the ultimate and practical solution of the challenge. Although several such methods have been suggested[1-8], exploring and producing multiple reaction pathways of a complex system remains a challenge. The objective of this work is to present a method that can efficiently explore and produce multiple reaction pathways connecting two end states. Other approaches using a conformational driving force do not sample alternatives[9,10]. Methods that are robust, such as transition path sampling[2,3], are very expensive to use for complex systems in the presence of multiple steps and barriers.

Various chain-of-states methods have been suggested based on the assumption that a dominant transition pathway between two states follows the minimum energy pathway[11-13]. The limitations of these methods are that they do not consider the dynamics of a system and find only the nearest local minimum solution from a given initial pathway[1,9]. Alternative methods based on the principle of least action have been suggested[5,14-20]. Passerone and Parrinello suggested the action-derived molecular dynamics (ADMD) method based on the combination of classical action and a penalty term that conserves the total energy of a system[18,19]. To enhance the convergence of ADMD calculations, Lee et al.[20-23] introduced a kinetic energy penalty term based on the equipartition theorem. Although the ADMD approaches yield physically relevant pathways, they have two practical limitations[20,24]: (a) they strongly depend on the initial guesses of a pathway; and (b) they cannot identify the relative dominance of multiple pathways because the classical principle of least action is an extremum principle[25].

For diffusive processes, the second problem can be avoided by using the Onsager-Machlup (OM) action $S_{OM}$[15,26-31]. Onsager and Machlup showed that the relative probability to observe a pathway with an OM action of $S$ is proportional to $e^{-S/k_B T}$, where $k_B$ is the Boltzmann constant and $T$ is a temperature. Thus the most dominant pathway corresponds to the one that minimizes $S_{OM}$ and the same result can be obtained by solving the Fokker–Planck equation[7,8,32]. This property recasts the problem of finding multiple pathways into a global search and optimization problem. However, finding multiple low action pathways is a challenging task because the minimization of $S_{OM}$ requires the second derivatives of a potential function, which are computationally expensive, at best, and wholly unavailable for many quantum mechanical energy surfaces.

In this work, we propose an efficient computational method, Action-CSA, that finds multiple low OM action pathways without second derivative calculations. For global search and optimization of a pathway space, we used an efficient global optimization method called conformational space annealing (CSA), which is based on a combination of genetic algorithm, simulated annealing, and Monte Carlo with minimization[33,34]. CSA has been demonstrated to be extremely efficient in solving various global optimization problems including finding low energy conformations of Lennard–Jones clusters[35], protein structure prediction[34-40], community detection in networks[41-43], and designing the first-ever direct bandgap silicon and carbon allotropes[44-47]. CSA is the most robust method available in CHARMM[48,49] for generating low energy conformations of peptides. We extend the CSA approach to examine pathways, preserving all features that make it robust and efficient, by applying it to sets of entire pathways represented as a chain-of-states.

Action-CSA efficiently explores the pathway space regardless of the heights of energy barriers via crossovers and mutations of pathways. Without calculating the second derivatives of a potential energy, multiple diverse pathways with low OM action were obtained by combining local optimization of pathways using classical action and selection of pathways using the OM action. From benchmark simulations using alanine dipeptide, our method finds multiple transition pathways, which are consistent with long-time Langevin dynamics (LD) simulations. The rank order statistics and transition time distributions of the multiple pathways are in good agreement with those of the LD results. For the conformational change of hexane from the all-gauche( − ) to all-gauche( + ) states, Action-CSA finds all possible transition pathways. Also, the lowest action folding pathway of FSD-1 is consistent with recent experiments reported after the submission of this work. These results demonstrate that Action-CSA searches multiple reaction pathways including the most dominant one in an efficient and robust way.

## Results

**Conformational change of alanine dipeptide.** A comparison of Action-CSA and LD simulations demonstrate that Action-CSA finds multiple possible pathways and correctly identifies the most probable one. Eight different pathways were identified for the $C7_{eq} \rightarrow C7_{ax}$ transition by clustering all pathways sampled from the Action-CSA simulations (Fig. 1a). From the $S_{OM}$ values obtained with different transition times (Fig. 1b), it is clear that the pathway that crosses barrier B has the lowest $S_{OM}$ value along all transition times tested indicating that it is the most probable pathway regardless of the transition time. This is consistent with the 500 μs LD simulation results (Table 1). From the LD simulations, 1,350 transitions starting from $C7_{eq}$ to $C7_{ax}$ were observed. They were clustered by finding the nearest neighbor from the eight pathways obtained by Action-CSA. From the clustering, the pathway crossing barrier B was identified as the most probable one with all transition times considered. This demonstrates that Action-CSA correctly identified the minimum OM action pathway and that it matched the most dominant pathway observed in the LD simulations.

In addition, it is also identified that Action-CSA simulations provide information on the transition times of various pathways. Until $t < 0.8$ ps, the pathway that crosses barrier C (Path2) has the second lowest $S_{OM}$ and the lowest $S_{OM}$ value was observed at 0.4 ps. These are consistent with the LD results in which all 118 transitions that crossed barrier C occurred within 1.1 ps and their most probable transition time was 0.7 ps (the inset of Fig. 1b). However, when $t > 0.8$ ps, Path3, which passes the fully extended conformation region $(\Phi, \Psi) = ( − 180°, 180°)$ and barrier A and B becomes the pathway with the second lowest $S_{OM}$. From the LD simulations, when $t > 0.9$ ps, 25 pathways similar to Path3 were identified, which makes them the second dominant pathway. These results demonstrate that the profile of $S_{OM}$ values is consistent with the distributions of transition times obtained from the LD simulations. Note that the most probable transition times observed from the LD simulations are longer than the minimum action transition times obtained from the CSA simulations. This is because high-frequency motions due to

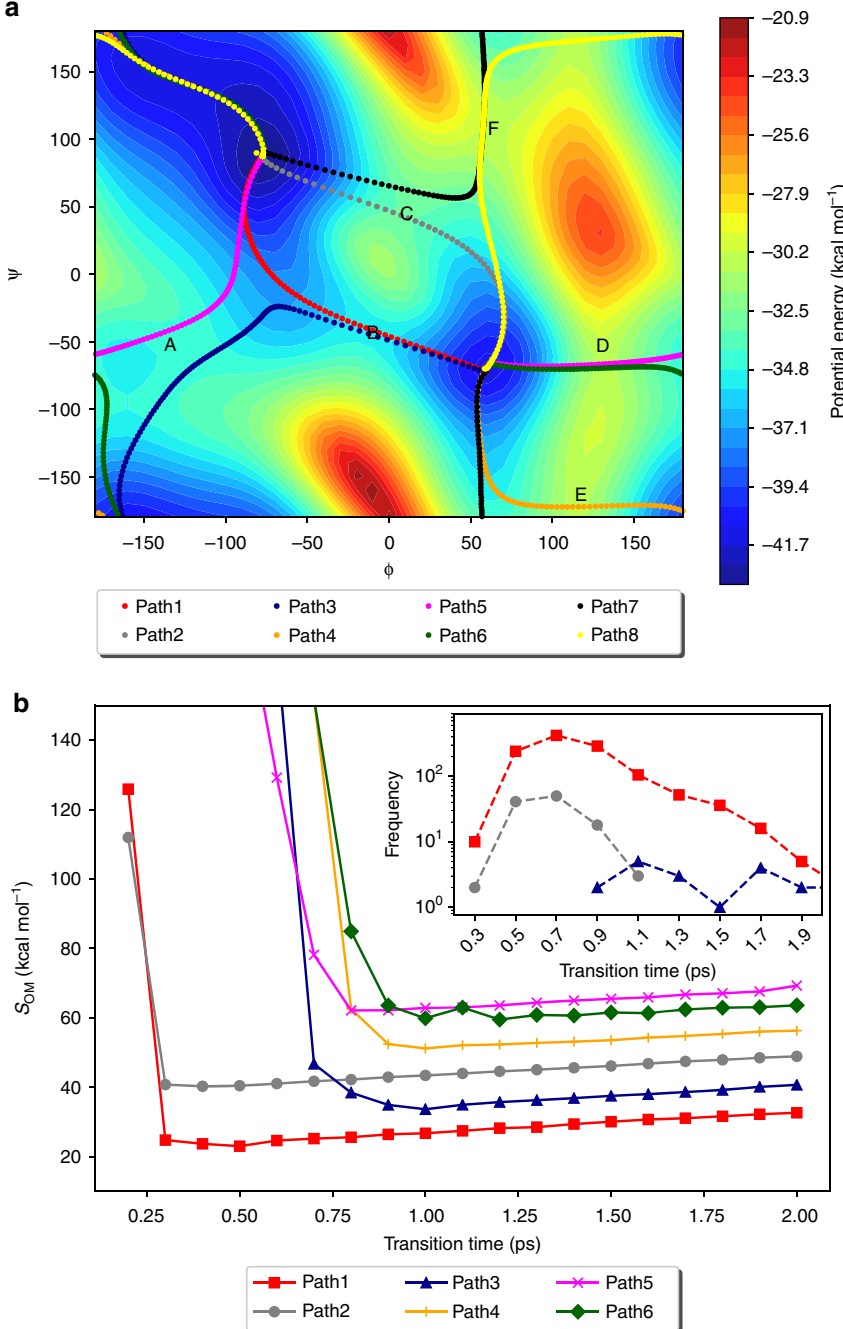

**Figure 1 | Conformational transition pathways of alanine dipeptide. (a)** Eight different pathways for the $C7_{eq} \rightarrow C7_{ax}$ transition selected by OM action values and the potential energy surface for the $\Phi$ and $\Psi$ angles with the PARAM19 force field (in units of kcal mol$^{-1}$) are shown. Potential energy barriers are labelled in order of their heights (from A to F). **(b)** The $S_{OM}$ values of six pathways for the $C7_{eq} \rightarrow C7_{ax}$ transitions of alanine dipeptide along different transition times are shown.

thermal fluctuations are filtered out in the minimum action pathways[1,15,16]. This means that the dwell time is well filtered out in the simulation, where a physically sufficient sampling time is assumed.

**Conformational transition of hexane.** The second example is finding multiple low-lying pathways for the conformational change of hexane from the all-gauche($-$) state (g$-$g$-$g$-$) to the all-gauche($+$) state (g$+$g$+$g$+$). We assessed the sampling ability of Action-CSA by investigating the diversity of sampled pathways. If it is assumed that dihedral angles do not cross the

highest energy barrier around the *cis* state, all possible transition pathways can be enumerated (Supplementary Table 2). For the transition under this assumption, there exist 44 possible pathways in total, excluding cases where a torsional barrier is crossed multiple times. If the symmetries of dihedral angles and the atomic order are considered, these 44 pathways can be reduced to 14 pathway types.

We repeated the Action-CSA calculation of the transition 40 times by using 200 initial pathways consisting of 100 replicas and a transition time of 3 ps. In all 40 simulations, the 6 lowest action pathways, CC$+$, CC$-$, TC$+$, TC$-$, CM$+$ and CM$-$, were found in a robust fashion. The highest-action pathway, MXM,

was found in nine simulations, and the other seven higher-action pathways were found in at least 29 simulations. On average, a single CSA simulation sampled 12 out of 14 unique path types and 26 out of 44 possible pathways. These results show that Action-CSA can sample a number of lowest action pathways including the most dominant one. The majority of the remaining pathways with higher actions can also be found with a tendency that lower action-value pathways are more frequently found. We note that the sampling ability of Action-CSA can be further improved by increasing the bank size. The potential energy landscape of the CC+ pathway corresponding to the least $S_{OM}$ shows that hexane crosses six energy barriers (Fig. 2). It should be noted that the fraction of possible pathways found in a given Action-CSA simulation depends on the number of replicas and the transition time. This example represents typical use, and not a best case scenario.

**Folding pathway of mini protein FSD-1.** The third example is finding the folding pathway of FSD-1, a 28-residue mini-protein that has been widely investigated as a model system for studying the protein folding problem[22,50–54]. Folding pathways of FSD-1 from the fully extended conformation to the native structure were represented by using 100 replicas, a total folding time of 10 ps, and a temperature of 300 K. The protein was represented by the PARAM19 force field[55] and solvation effects were considered by the FACTS implicit solvent model[56]. This calculation required

approximately 160 h with 72 Haswell cores and a diverse set of about twenty low action pathways were generated.

The lowest OM action folding pathway is consistent with a recent experiment[57] published after the submission of this work, where the early formation of C-terminal α-helix is observed to be followed by the concurrent formation of the β-hairpin and hydrophobic contacts. A comparison of the root mean square deviation values indicates that the α-helix approaches to the native structure earlier than the β-hairpin. Afterward, the folding of β-hairpin and the formation of hydrophobic core occur concurrently (Fig. 3a). The potential energy landscape of the FSD-1 folding shows that the potential energy decreases quickly after the 80 step suggesting that this step may be the transition state of folding (Fig. 3b). The conformation at the 80 step shows that the α-helix is almost fully formed while the C-terminal region is not folded yet and the hydrophobic core is partially exposed.

After our manuscript was submitted, it came to our attention that Meuzelaar and co-workers reported that the folding of FSD-1 occurs via an intermediate state where only the α-helix is formed[57]. After this intermediate state, the β-hairpin and hydrophobic contacts form. The pathway was determined by combining temperature-dependent UV circular dichroism, Fourier transform infrared spectroscopy, two-dimensional infrared spectroscopy, and temperature-jump transient-IR spectroscopy. This folding mechanism shows good agreement with the dominant folding pathway identified in this study. This agreement strongly indicates that our method can serve as a powerful tool to study the folding mechanism of a protein with

**Table 1 | The frequencies of transition pathways of alanine dipeptide from $C7_{eq}$ to $C7_{ax}$ observed from 500 μs Langevin dynamics simulations.**

| Path ID | Frequency |
|---------|-----------|
| Path1 | 1,183 |
| Path2 | 116 |
| Path3 | 25 |
| Path4 | 7 |
| Path5 | 4 |
| Path6 | 4 |
| Path7 | 10 |
| Path8 | 1 |

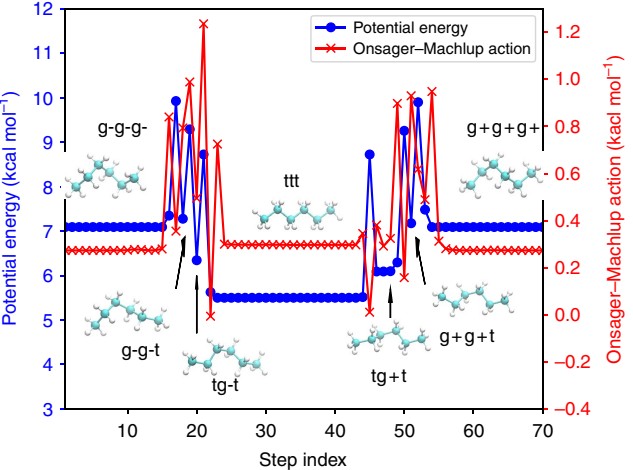

**Figure 2 | The lowest OM action pathway of conformational transition of hexane.** The changes of potential energy and Onsager–Machlup action along the lowest action pathway between the all-gauche(−) to the all-gauche(+) conformations of hexane in the vacuum, the CC+ pathway, are illustrated.

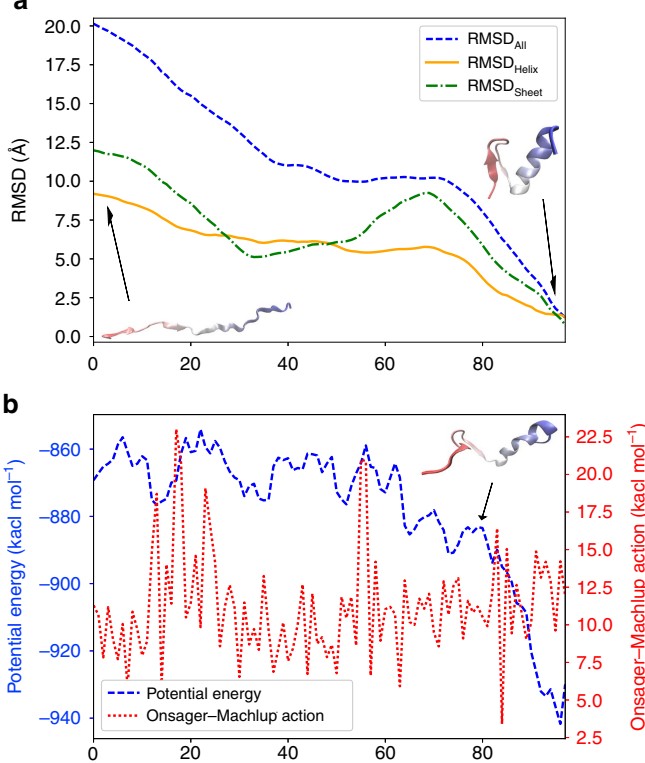

**Figure 3 | The lowest action folding pathway of FSD-1. (a)** The root mean square deviation values of the entire FSD-1 (blue), the C-terminal α-helix (green, residue 14–28), and the N-terminal β-hairpin (orange, residue 1–13) from the native structure along the folding pathway are displayed. **(b)** The evolutions of potential energy (blue) and the Onsager–Machlup action (red) of FSD-1 along the folding pathway are displayed.

atomic details. We note that additional sub-optimal folding pathways were also obtained, where the second lowest OM action pathway suggested a different pathway: the β-hairpin folds first, followed by the concurrent formation of α-helix and hydrophobic contacts (Supplementary Fig. 1).

## Discussion

The goal of Action-CSA is to search multiple diverse pathways of low OM action values in a fast and efficient fashion, rather than sampling a specific physical ensemble. Throughout an Action-CSA calculation, the OM action is used to identify the relative probabilities of multiple trial pathways obtained by performing crossover and mutation operations followed by local minimization using the classical action. Ideally, we should have performed the local minimization using the OM action, which was not feasible due to the high cost of calculating second derivatives. Instead, we performed one-point evaluation of the OM action after the minimization. Here we assumed that the CSA selection procedure using the OM action drives the CSA population to low OM action basins. A similar approach was used in designing the first-ever direct bandgap silicon and carbon allotropes[44–47], where the local optimization was performed in terms of enthalpy but the selection was done by the band gap property. Thus, an Action-CSA calculation yields a set of low OM action pathways, but they do not correspond to any physical ensembles. Action-CSA results can be used as the starting point for existing pathway sampling methods, such as transition pathway sampling[3] or the replica-exchange pathway sampling[30,31], which aim to generate canonical ensembles. In addition, our method can be used to find low potential energy pathways or multiple Newtonian pathways via selection using the height of potential energy barrier or the Gauss action[58,59] instead of the OM action.

We note that the OM action depends on the friction parameter. As friction approaches to zero, the Langevin dynamics, described by the OM action, converges to the Newtonian dynamics, described by the classical action. Thus, when friction is small, we expect that pathways sampled with the classical action will be close to those sampled with the OM action. However, when friction is large, this assumption may not hold. In such cases, one should minimize the OM action directly, which will be computationally much more expensive than the current scheme because the analytic gradients of the OM action require Hessian calculations (equation (4)).

Action-CSA has three unique characteristics compared with existing path sampling methods[2–4,29,60–62]: (a) the use of the bank of diverse pathways, (b) the generation of new trial pathways by swapping and mutating of pathway segments followed by local optimization using classical action with total energy restraint, and (c) the selection of pathways with higher probabilities by using the OM action. By maintaining a diverse bank population, one can perform more extensive search of the pathway space, and is less reliant on the initial pathway chosen. The crossover and mutations of pathways followed by local optimization also facilitate extensive search of pathway space because those operations generate new pathways by overcoming large energy barriers, which is the major limitation of MD-based approaches. Last, the combined use of the classical action for local minimization and the OM action for selection is computationally relatively efficient, and allowed us to find multiple low OM action pathways without performing the computationally expensive Hessian calculation. Combined, this computational efficiency thus allows investigation of larger/more complicated systems.

Elber et al.[16] optimized the Gauss action using simulated annealing (SA) to find the folding pathway of C-peptide, which is a 16-residue long peptide forming a helical conformation. The

sampling efficiency of SA depends on its annealing schedule, the maximum temperature set during the annealing, and the heights of energy barriers. Since SA is based on molecular dynamics, which is history-dependent, the probability to find the global minimum of a system depends on the initial state. Therefore, it may take enormously long time to sample the entire pathway space when the degrees of freedom are large, and/or the energy landscape is highly rugged. In addition to these limitations, SA using the Gauss action requires computationally expensive Hessian calculation of the potential energy. In the work by Fujisaki et al.[30,31], the ensemble of pathways was sampled using the replica exchange molecular dynamics (REMD) and the OM action. Although REMD is known to be superior to SA in terms of its sampling efficiency, it also suffers from similar limitations of SA. Due to these limitations, only relatively simple model systems, Bolhuis' two-dimensional potential[63] and a coarse-grained self-avoiding polymer with one bead type and three interaction terms[64], were studied.

In conclusion, we demonstrated that efficient global optimization of Onsager-Machlup action reveals multiple transition pathways including the most dominant one successfully. In this work, we introduced a computational method that samples multiple possible pathways and provides information on the relative dominance of them via efficient global optimization of Onsager-Machlup action using the CSA method. The advantages of our method over existing pathway sampling methods are in the fact that its sampling efficiency is independent of the quality of initial guesses on pathways; only the calculation of first derivatives is required; and its sampling ability is not limited by the existence of high energy barriers separating pathways, which is a major limiting factor of previous MD-based pathway sampling methods in exploring the pathway space[7,8,16,28,30–32]. Also, it is identified that the profile of minimum Onsager-Machlup actions found with different transition time parameters provides kinetic information on multiple pathways. In terms of implementation, Action-CSA calculation is massively parallel because the local optimization of each trial pathway is independent of each other. Thus, pathway samplings for larger systems are possible with the help of a large computer cluster system. We anticipate that the Action-CSA method will be used as a first-step exploration for complex reactions and large-scale conformational changes due to its low cost and robust nature. Results from Action-CSA can be used as the starting point for many other methods.

## Methods

**Classical and Onsager-Machlup actions.** Here, we briefly review the theoretical background behind Action-CSA. If a system with $N$ atoms with a potential energy $V$ follows the overdamped Langevin dynamics,

$$\gamma\dot{\mathbf{x}} = -\frac{\partial V}{\partial \mathbf{x}} + \mathbf{R}, \tag{1}$$

where $\mathbf{x}$ is a $3N$ dimensional mass-weighted coordinate vector, $\gamma$ is collision frequency, and $\mathbf{R}$ is a Gaussian random force, the relative probability of finding a final state $\mathbf{x}_f$ at a time $t$ from an initial state $\mathbf{x}_i$ via diffusive trajectories $\mathbf{x}(t')$ is determined by using the path integral approach and OM action $S_{OM}[\mathbf{x}(t')]$[26,27]:

$$P(\mathbf{x}_f|\mathbf{x}_i; t) = \int_{\mathbf{x}(0)=\mathbf{x}_i}^{\mathbf{x}(t)=\mathbf{x}_f} \mathcal{D}\mathbf{x}(t') \exp\left(-\frac{S_{OM}[\mathbf{x}(t')]}{k_B T}\right), \tag{2}$$

where $\mathcal{D}\mathbf{x}(t')$ indicates that the integration runs over all possible pathways $\mathbf{x}(t')$. This relationship suggests that if the $S_{OM}$ values of all physically accessible pathways are obtained, one can determine the relative populations of multiple pathways. Thus, $S_{OM}$ is a proper target objective function of global optimization. The generalized OM action of a pathway $\mathbf{x}(t)$ is defined[26,27,65,66]:

$$S_{OM}[\mathbf{x}(t)] = \frac{\Delta V}{2} + \frac{1}{4\gamma}\int_0^t d\tau\{[\gamma\dot{\mathbf{x}}(\tau)]^2 + |\nabla V[\mathbf{x}(\tau)]|^2 - 2k_B T\nabla^2 V[\mathbf{x}(\tau)]\}, \tag{3}$$

where $\Delta V = V(\mathbf{x}_f) - V(\mathbf{x}_i)$. In the original formula of action derived by Onsager and Machlup, the last term of equation (3) was absent[26,27]. It was shown that, for

the purposes of reweighting and sampling diffusive pathways, two OM actions with and without the Hessian term are equivalent. However, for the purpose of finding the most probable trajectory motif, the term should be considered because it represents the entropic corrections connected with fluctuations and the neighborhood of a given trajectory motif, which is also represented as a tube around the motif[66,67]. Note that the minimization of $S_{OM}$ using analytic local minimization algorithms requires analytic third derivatives. This makes the direct global optimization of $S_{OM}$ hard to be applied to detect transition pathways of biomolecules with all-atom force fields due to the complexity of implementation and high computational cost. For numerical calculations based on a chain-of-states representation, the OM action should be discretized. The method uses the second-order discretization of the symmetric OM formula, which uses only gradients for $S_{OM}$ calculations[68]:

$$S_{OM}[\mathbf{x}(t)] = \frac{\Delta V}{2} + \sum_{i=0}^{P-1} \frac{\Delta t}{4\gamma} \left\{ \left[ \frac{\gamma(\mathbf{x}_{i+1} - \mathbf{x}_i)}{\Delta t} \right]^2 + \frac{|\nabla V(\mathbf{x}_i)|^2 + |\nabla V(\mathbf{x}_{i+1})|^2}{2} \right. $$
$$\left. - \frac{\gamma(\mathbf{x}_{i+1} - \mathbf{x}_i)}{\Delta t} \cdot [\nabla V(\mathbf{x}_{i+1}) - \nabla V(\mathbf{x}_i)] \right\}, \quad (4)$$

where $P+1$ is the number of replicas, $\Delta t$ is a time step between successive replicas, and $t = P\Delta t$ is the total transition time. This formula is more efficient than the direct implementation of equation (3) since it requires only the first derivatives of $V$ to evaluate $S_{OM}$.

**Global action optimization.** Here, we describe the application of CSA to optimize $S_{OM}$. In general, a pathway is represented as a chain of $P-1$ replicas with $N$ atoms for each replica leading to $3N(P-1)$ total degrees of freedom. Each replica is represented by a sequence of $3N-6$ internal dihedral angles and 6 net translational/rotational degrees of freedom. An Action-CSA calculation starts with a set of random pathways on a pathway space. Subsequently, the actions of the random pathways are locally optimized.

As stated previously, direct minimization of $S_{OM}$ using analytic gradients is computationally challenging. For a computationally feasible local action optimization, we optimized a pathway using a modified action from ADMD instead of using $S_{OM}$. The discretized classical action is defined:

$$S_{classical}[\mathbf{x}(t)] = \sum_{i=0}^{P-1} L_i(\mathbf{x}_i)\Delta t = \sum_{i=0}^{P-1} \left[ \frac{(\mathbf{x}_i - \mathbf{x}_{i+1})^2}{2\Delta t^2} - V(\mathbf{x}_i) \right] \Delta t. \quad (5)$$

Physically accessible pathways correspond to the stationary points of $S_{classical}$. Finding such pathways is a computationally difficult task because $S_{classical}$ is not bounded; $S_{classical}$ can be minimized or maximized, and the stationary points of $S_{classical}$ can be minima, maxima or saddle points. Another practical problem is that the total energies of pathways satisfying the stationary condition $\delta S_{classical} = 0$ may not be conserved[18]. To find pathways that satisfy the principle of least action and conserve total energies, a modified action with a penalty term restraining total energy was suggested[18]:

$$\Theta(\mathbf{x}_i; E) = \mu_A S_{classical} + \mu_E \sum_{i=0}^{P-1} (E_i - E)^2$$
$$= \mu_A \sum_{i=0}^{P-1} \left[ \frac{(\mathbf{x}_i - \mathbf{x}_{i+1})^2}{2\Delta t^2} - V(\mathbf{x}_i) \right] \Delta t + \mu_E \sum_{i=0}^{P-1} \left\{ \left[ \frac{(\mathbf{x}_i - \mathbf{x}_{i+1})^2}{2\Delta t^2} + V(\mathbf{x}_i) \right] - E \right\}^2, \quad (6)$$

where $E$ is a targeted total energy of a system, $\mu_A$ and $\mu_E$ are the weighting parameters of the classical action, and the restraint term for energy conservation. The minimization of $\Theta[\mathbf{x}(t); E]$ requires only the first derivatives of $V$.

The set of locally optimized initial random pathways using $\Theta[\mathbf{x}(t); E]$ is called the *first bank*. The first bank remains the same throughout the optimization and is used as the reservoir of partially optimized pathways to enhance the diversity of pathway search. A copy of the first bank is generated and called a *bank*. The pathways in the bank are updated during a calculation while the size of the bank is kept constant. By using the pathways included in the first bank and the bank, new trial pathways are generated by crossover and mutation (random perturbation) operations. For a crossover operation, two pathways, a seed pathway from the bank and a random pathway either from the bank or the first bank, are selected and random parts of two selected pathways are swapped. For a random perturbation, a certain number of degrees of freedom of a seed pathway, up to 5% of total degrees of freedom, are randomly changed. The generated trial pathways are locally optimized by minimizing $\Theta[\mathbf{x}(t); E]$ to remove any possible artifacts generated by the crossover and the mutation operations. After local optimizations, the bank is updated by comparing the $S_{OM}$ values of the existing pathways and the new ones instead of $\Theta[\mathbf{x}(t); E]$.

A key feature of CSA is a sophisticated bank-update procedure that prevents a search being trapped in local minima during the optimization and keeps the diversity of the bank. For a newly obtained configuration, a pathway in this work, $\alpha$, the pathway separation distances $D$ between $\alpha$ and the existing ones in the bank are calculated. If the distance between $\alpha$ and its closest neighbor is less than a cutoff distance $D_{cut}$, only the better configuration in terms of the objective function, $S_{OM}$ in this work, is selected. If $D > D_{cut}$, $\alpha$ is considered a new configuration and it

replaces the worst configuration in the bank if it is better. At initial stages of a calculation, $D_{cut}$ is kept large for wider sampling. As the calculation proceeds, it gradually decreases for a refined search near the global minimum. The bank keeps updating until no better configuration is found. In this work, a distance between two pathways was measured by the Fréchet distance[69]. More details on a general CSA procedure are described elsewhere[33–35,37,39,40].

**Action-CSA simulation.** To verify that Action-CSA successfully finds multiple pathways and allows one to determine the rank order of the pathways based on their optimized $S_{OM}$ values, we applied our method to investigate the conformational transition of alanine dipeptide from $C7_{eq}$ to $C7_{ax}$ in the vacuum. Here, we used the polar hydrogen representation in the PARAM19 force field[55] and the dielectric constant was set to 1.0 (ref. 70). We performed Action-CSA simulations with various transition times, $t$ in equations (4) and (6), ranging from 0.2 to 2.0 ps with an interval of 0.1 ps. The numbers of replicas were adjusted with $t$ to keep the time step between successive replicas $\Delta t = 5$ fs. All simulations were performed at temperature $T = 350$ K with a collision frequency $\gamma = 1.0$ ps$^{-1}$. The reference total energy $E$ in equation (5) was obtained by adding the initial potential energy $V(\mathbf{x}_i) = -43.3$ kcal mol$^{-1}$ and a kinetic energy of 12.5 kcal mol$^{-1}$ estimated by $3Nk_BT/2$ with the number of atoms $N = 12$. The weighting parameters $\mu_A$ and $\mu_E$ in equation (5) were set to $-1.0$ and 1.0, respectively. For comparison purposes, we performed 5,000 independent 100 ns LD simulations of alanine dipeptide under the same condition amounting to 500 $\mu$s LD simulations and counted the number of the $C7_{eq} \rightarrow C7_{ax}$ transitions. An Action-CSA calculation requires 10 adjustable parameters, and they are listed in Supplementary Table 1. The parameters for the calculations presented in this study were not extensively optimized. Rigorous optimization of the parameters is out of the scope of this study, and requires a series of subsequent benchmark studies.

**Data availability.** The Action-CSA code is freely available for academic, government and nonprofit use as a part of the CHARMM molecular dynamics package (http://charmm.chemistry.harvard.edu/). All relevant data are available from the authors upon request.

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

## Acknowledgements

We acknowledge helpful discussions with Attila Szabo and Richard Pastor. The authors wish to acknowledge Steven Gross for his critical reading of the manuscript. Juyong Lee and B.R.B. were supported by the Intramural Research Program of the NIH, NHLBI under Project No. Z01 HL001051-20. I.-H.L., I.J. and Jooyoung Lee were supported by the National Research Foundation of Korea (NRF) under Grant No. 2008-0061987 funded by the Korea government (MEST). I.-H.L. was also supported by Samsung Science and Technology Foundation under Grant No. SSTF-BA1401-08. Computational resources and services used in this work were provided by the LoBoS cluster of the National Institutes of Health.

## Author contributions

Juyong Lee conceived and designed the study, designed and implemented the Action-CSA algorithm, performed the simulations, analysed the results and wrote the paper. I.-H.L. contributed to the implementation of the Action-CSA algorithm. I.J. contributed to the design of the Action-CSA algorithm. Jooyoung Lee supervised the study, contributed to the design of the Action-CSA algorithm, analysed the results and wrote the paper. B.R.B. supervised the study, contributed to the design of the Action-CSA algorithm, analysed the results and wrote the paper. All authors were involved in manuscript editing.

## Additional information

**Competing interests:** The authors declare no competing financial interests.

**Publisher's note**: 

