## [Peer Review File · Nature Communications]

Reviewers' comments:

Reviewer #1 (Remarks to the Author):

The problem of global searches for pathways is a long-standing challenge in computational kinetics. Even the optimization of a single structure (e.g. the protein folding problem) is a far from trivial task. The problem of finding an optimal path is expected to be even harder. Lee et al. proposed a two-step mechanism for global path optimization. In the first step they optimize a classical action with an additional constraint of a constant energy. They use a clever path optimization algorithm that allows for swapping of path segments and reoptimization. It sounds like one of the more sophisticated approaches for path optimization that I encountered.

In the second step they use the Onsager Machlup action to decide on path weight. This is similar in spirit to the approach taken by Facioli using ratchet trajectories and then re-weighting them by the OM action. Do the authors re-weight their paths to switch between ensembles? The author should comment on that as I am puzzled by the issue of path weights. What is the path ensemble that the authors aim to generate? Are those paths in the microcanonical or canonical ensembles? Or do they just want to identify minimum energy pathways? The classical action with energy constraint are, I believe, in the microcanonical ensemble, while the OM paths are in the canonical ensemble. How the different ensembles are brought together is not clear. One also notes that the OM action depends on the friction parameter while the classical action is not. The resulting OM paths will depend on the friction value at finite temperatures. Again the connection to the classical action is not obvious to this reviewer.

Minor points

1. ADMD is not defined.
2. I am unclear what are some of the different pathways in Fig. 1. For example, there are yellow and black paths near the letter C. Are they results of optimal OM action, classical action path, minimum energy path?

In summary, I find the ability of globally optimizing pathways intriguing and important. Such an algorithm will have a significant impact on the field. Overall I am enthusiastic about this paper. Nevertheless, it will be useful to clarify the above points to make the paper clearer and more usable by the community of researchers in the field.

Reviewer #2 (Remarks to the Author):

The authors present a path-finding method based on the Onsager-Machlup (OM) action and apply it to simple systems and a protein-folding system. Although the work is quite interesting and carefully done, I am not convinced the authors have achieved as much of an advance as they claim, for several reasons: (1) Qualitatively, their work is similar to prior efforts and similar theory is used as the basis. (2) A number of existing path sampling methods are dismissed as impractical, but they have been applied to systems that appear equally challenging. (3) The claim that the dominant pathway found for the folding system is well validated seems shaky.

(1) What distinguishes the method from related OM/action methods, such as the work by Elber and coworkers [ref 11] which was applied to folding? What about the work by Fujisaki and coworkers?

(2) The authors claim other (non-action) methods are not practical. I am aware of a number of path sampling methods (transition path sampling, transition interface sampling, milestoning, weighted ensemble, dynamic importance sampling, forward flux sampling), and most if not all have been applied to challenging protein problems. At least some of these methods are truly parallel. What is really the unique capability of the new method compared with existing approaches?

(3) Is the folding pathway found really validated by experiment? The fact that the “N-terminal-hairpin is more exible than the C-terminal-helix” in experiment is very indirect confirmation of the simulation data. Comparison with MD is helpful, but this undercuts the claim that the new method is uniquely powerful. Overall, the data presented are underwhelming.

Additional technical questions:

Is entropy of paths accounted for in optimization scheme?

What is importance of time steps and how was it checked?

What is the sensitivity of the method to the weighting parameters of Eq (6)?

Reviewer #3 (Remarks to the Author):

The authors present a global search strategy for finding multiple reaction pathways that are founded on a maximization of the Onsager-Machlup (OM) action. They present a new way to apply the OM formula in its discretized way in combination with CSA (Conformational space annealing). CSA is used to optimize the OM action. Therefore, only the first derivatives with respect to the potential energy are needed. This is an improvement over older implementations, which used second derivatives. (related to formula 3 in the manuscript). Related to this, Fujisaki et al. (J. Chem. Phys. 132, 134101 (2010)) presented some work by using the replica exchange approach for the Bolhuis potential. The authors present an applicable tool for applying it to real chemistry problems. Their bank approach seems to be indeed an intelligent way to guarantee the diversity within the whole procedure. For all their tests, they present a trustworthy proof either computationally through Langevin Dynamic simulations or experiments.

In this way, the paper seems to be an improvement over existing approaches and is recommended for publication.

Related to equation 1 one question remains open. Would it be the better choice to introduce mass weighted coordinates directly? For Brownian dynamics (overdamped Langevin dynamics) the second derivatives can be neglected if the mass goes to zero, then the direct use of m would be replaced and already be included in x .

Reviewer #1 (Remarks to the Author):

The problem of global searches for pathways is a long-standing challenge in computational kinetics. Even the optimization of a single structure (e.g. the protein folding problem) is far from trivial task. The problem of finding an optimal path is expected to be even harder. Lee et al. proposed a two-step mechanism for global path optimization. In the first step, they optimize a classical action with an additional constraint of constant energy. They use a clever path optimization algorithm that allows for swapping of path segments and reoptimization. It sounds like one of the more sophisticated approaches for path optimization that I encountered. In the second step, they use the Onsager Machlup action to decide on path weight. This is similar in spirit to the approach taken by Facioli using ratchet trajectories and then re-weighting them by the OM action.

Note: Below, in response to Reviewer #1, we first provide our answers to the reviewer's questions, and at the end summarize how we have revised the manuscript to address these questions for new readers.

Do the authors re-weight their paths to switch between ensembles?

We don't perform any reweighting procedures during the CSA optimization of the OM action. The search by CSA does not correspond to any ensembles. Only after low OM action pathways are obtained by Action-CSA, we reweight physical pathways.

The author should comment on that as I am puzzled by the issue of path weights. What is the path ensemble that the authors aim to generate?

The goal of Action-CSA method is to search multiple diverse pathways with low OM action values in an efficient fashion, rather than sampling a specific physical ensemble.

Are those paths in the microcanonical or canonical ensembles?

A set of pathways obtained with Action-CSA does not correspond to any physical ensembles. It is a set of diverse lowest OM action pathways of a given reaction.

Or do they just want to identify minimum energy pathways?

Throughout our study, we produced no minimum energy pathways. All pathways reported in the manuscript correspond to low OM action pathways.

The classical action with energy constraint are, I believe, in the microcanonical ensemble, while the OM paths are in the canonical ensemble. How the different ensembles are brought together is not clear.

The reviewer is absolutely right about the first sentence. The reason for using the classical action for local minimization of trial pathways was to find physically relevant pathways quickly without performing computationally expensive Hessian calculations. After the minimization, we performed one-point evaluation of the OM action, so that the CSA selection procedure using the OM action can drive the CSA population to low OM action basins. It should be noted that CSA performs searching not sampling.

One also notes that the OM action depends on the friction parameter while the classical action is not. The resulting OM paths will depend on the friction value at finite temperatures. Again, the connection to the classical action is not obvious to this reviewer.

Again the reviewer is correct here. As friction approaches to zero, the Langevin dynamics, described by the OM action, converges to the Newtonian dynamics, described by the classical action. Thus, when friction is small, we expect that pathways sampled with the classical action will be close to those sampled with the OM action.

Minor points

1. ADMD is not defined.

ADMD is the acronym of Passerone and Parrinello's action-derived molecular dynamics method. We believe that it was defined in the original manuscript. The term is now defined in line 46 on page 2 as follows: "Passerone and Parrinello suggested the action-derived molecular dynamics (ADMD) method based on the combination of classical action and a penalty term that conserves the total energy of a system [14, 15]."

2. I am unclear what are some of the different pathways in Fig. 1. For example, there are yellow and black paths near the letter C. Are they results of optimal OM action, classical action path, minimum energy path?

All paths discussed in this manuscript are low OM action pathways, and all paths shown in Figure 1 are low OM action pathways. To clarify this, the caption is revised from "eight different pathways for the C7eq \rightarrow C7ax transition" to "eight different pathways for the C7eq \rightarrow C7ax transition selected by OM action values".

In summary. I find the ability of globally optimizing pathways intriguing and important. Such an algorithm will have a significant impact on the field. Overall I am enthusiastic about this paper. Nevertheless, it will be useful to clarify the above points to make the paper clearer and more usable by the community of researchers in the field.

We greatly appreciate the reviewer for the insightful comments, which helped us to improve the presentation of our manuscript. To clarify the ambiguity issues raised by the reviewer, especially on what can be obtained by Action-CSA and how various actions are connected to each other, we added the following paragraphs on page 9:

"The goal of Action-CSA is to search multiple diverse pathways of low OM action values in a fast and efficient fashion, rather than sampling a specific physical ensemble. Throughout an Action-CSA calculation, the OM action is used to identify the relative probabilities of multiple trial pathways obtained by performing crossover and mutation operations followed by local minimization using the classical action. Ideally, we should have performed the local minimization using the OM action, which was not feasible due to the high cost of calculating second derivatives. Instead, we performed one-point evaluation of the OM action after the minimization. Here we assumed that the CSA selection procedure using the OM action drives the CSA population to low OM action basins. A similar approach was used in designing the first-ever direct bandgap silicon and carbon allotropes [43–46], where the local optimization was performed in terms of enthalpy but the selection was done by the band gap property. Thus, an Action-CSA calculation yields a set of low OM action pathways, but they do not correspond to any physical ensembles. Action-CSA results can be used as the starting point for existing pathway sampling methods, such as transition pathway sampling [5] or the replica-exchange pathway sampling [26, 27],

which aim to generate canonical ensembles. In addition, our method can be used to find low potential energy pathways or multiple Newtonian pathways via selection using the height of potential energy barrier or the Gauss action [57, 58] instead of the OM action.

We note that the OM action depends on the friction parameter. As friction approaches to zero, the Langevin dynamics, described by the OM action, converges to the Newtonian dynamics, described by the classical action. Thus, when friction is small, we expect that pathways sampled with the classical action will be close to those sampled with the OM action. However, when friction is large, this assumption may not hold. In such cases, one should minimize the OM action directly, which will be computationally much more expensive than the current scheme because the analytic gradients of the OM action require Hessian calculations (Eq. 4).”

=====
Reviewer #2 (Remarks to the Author):

The authors present a path-finding method based on the Onsager-Machlup (OM) action and apply it to simple systems and a protein-folding system. Although the work is quite interesting and carefully done, I am not convinced the authors have achieved as much of an advance as they claim, for several reasons: (1) Qualitatively, their work is similar to prior efforts and similar theory is used as the basis. (2) A number of existing path sampling methods are dismissed as impractical, but they have been applied to systems that appear equally challenging. (3) The claim that the dominant pathway found for the folding system is well validated seems shaky.

We appreciate the reviewer for the critical comments, which helped us to clarify and improve our manuscript significantly.

(1) What distinguishes the method from related OM/action methods, such as the work by Elber and co-workers [ref 11] which was applied to folding? What about the work by Fujisaki and co-workers?

In comparison with the work by Elber and co-workers [ref. 12], Action-CSA has three unique features: a) the use of the bank of diverse pathways, b) the generation of new trial pathways by swapping/crossover and mutating of pathway segments followed by local optimization using classical action with total energy restraint, and c) the selection of pathways with higher probabilities by using OM action. In the work by Elber and co-workers, the Gauss action was optimized by using the simulated annealing (SA) approach to find the folding pathway of the C-peptide. The sampling efficiency of SA heavily depends on its annealing schedule and the maximum temperature during the annealing, and the heights of energy barriers. In addition, since SA is based on molecular dynamics, which is history-dependent, the result of SA depends on the identity of the initial pathway. It would take impractically long time to sample the entire pathway space by using SA when the degrees of freedom are large, and/or the energy landscape is highly rugged. Also, to perform SA using the Gauss action, the Hessian of the potential energy should be calculated, which makes the method unsuitable for the application to complex systems due to its large computational burden. In the work by Fujisaki and co-workers, the ensemble of pathways was sampled using the replica exchange molecular dynamics (REMD) and the OM action. Although REMD is known to be superior to SA in terms of its sampling efficiency, REMD suffers from similar limitations of SA; its sampling efficiency depends on the maximum temperature and number of replicas, and the Hessian of a potential energy should be calculated. Due to these limitations, only relatively simple model systems, Bolhuis’ two-dimensional potential and a coarse-grained self-avoiding polymer with one bead type and three interaction terms, were amenable to study via these methods; in contrast, our method allows investigation of more complicated systems.

To be more specific on the uniqueness and strength of our method we cited more previous path sampling methods and added the following paragraphs as the second and third paragraphs of page 10:

“Action-CSA has three unique characteristics compared with existing path sampling methods [4, 5, 25, 59–62]: a) the use of the bank of diverse pathways, b) the generation of new trial pathways by swapping and mutating of pathway segments followed by local optimization using classical action with total energy restraint, and c) the selection of pathways with higher probabilities by using the OM action. By maintaining diverse bank population, one can perform more extensive search of the pathway space, and is less reliant on the initial pathway chosen. The crossover and mutations of pathways followed by local optimization also facilitate extensive search of pathway space because those operations generate new pathways by overcoming large energy barriers, which is the major limitation of MD-based approaches. Last, the combined use of the classical action for local minimization and the OM action for selection is computationally relatively efficient, and allowed us to find multiple low OM action pathways without performing the computationally expensive Hessian calculation. Combined, this computational efficiency thus allows investigation of larger/more complicated systems.

Elber and co-workers [11] optimized the Gauss action using simulated annealing (SA) to find the folding pathway of C-peptide, which is a 16-residue long peptide forming a helical conformation. The sampling efficiency of SA depends on its annealing schedule, the maximum temperature set during the annealing, and the heights of energy barriers. Since SA is based on molecular dynamics, which is history-dependent, the probability to find the global minimum of a system depends on the initial state. Therefore, it may take enormously long time to sample the entire pathway space when the degrees of freedom are large, and/or the energy landscape is highly rugged. In addition to these limitations, SA using the Gauss action requires computationally expensive Hessian calculation of the potential energy. In the work by Fujisaki and co-workers [26, 27], the ensemble of pathways was sampled using the replica exchange molecular dynamics (REMD) and the OM action. Although REMD is known to be superior to SA in terms of its sampling efficiency, REMD also suffers from similar limitations of SA and due to these limitations, only relatively simple model systems, Bolhuis’ two-dimensional potential [63] and a coarse-grained self-avoiding polymer with one bead type and three interaction terms [64], were studied.”

(2) The authors claim other (non-action) methods are not practical. I am aware of a number of path sampling methods (transition path sampling, transition interface sampling, milestoning, weighted ensemble, dynamic importance sampling, forward flux sampling), and most if not all have been applied to challenging protein problems. At least some of these methods are truly parallel. What is really the unique capability of the new method compared with existing approaches?

As in our reply to comment (1) above, the unique capabilities of Action-CSA are a) the use of the bank of diverse pathways, b) the generation of new trial pathways by swapping/cross-over and mutating of pathway segments followed by local optimization using classical action with total energy restraint, and c) the selection of pathways with higher probabilities by using OM action. The use of the bank enables extensive search of pathway space. The crossover and mutations of pathways followed by local minimization also facilitates extensive search of pathway space because those operations generate new pathways by overcoming large energy barriers, which is the major limitation of MD-based approaches. Last, the combined use of the classical action for local minimization and the OM action for selection allowed us to find multiple low OM action pathways without Hessian calculation. We addressed these characteristics in the paragraphs above. To further emphasize the characteristics, we added the following sentence at the last paragraph of page 3.

“Action-CSA can efficiently explore the pathway space regardless of the heights of energy barriers via crossovers and mutations of pathways. Without calculating the second derivatives of a potential energy, multiple diverse pathways with low OM action were obtained by combining local optimization of pathways using classical action and selection of pathways using the OM action.”

(3) Is the folding pathway found really validated by experiment? The fact that the “N-terminal-hairpin is more

flexible than the C-terminal-helix” in experiment is very indirect confirmation of the simulation data. Comparison with MD is helpful, but this undercuts the claim that the new method is uniquely powerful. Overall, the data presented are underwhelming.

During the revision, we found an experimental study (Meuzelaar et al., Folding of a Zinc-Finger $\beta\alpha$ -Motif Investigated Using Two- Dimensional and Time-Resolved Vibrational Spectroscopy, JPCB (2016), 120, 11151-11158), which was published after we submitted our manuscript to arXiv (arXiv:1610.02652). We were pleased to discover that the results of this study show excellent agreement with the most probable folding pathway identified in the manuscript. Based on this, we added the following paragraph as the last paragraph on page 8.

“After our manuscript was submitted, it came to our attention that Meuzelaar and co-workers reported that the folding of FSD-1 occurs via an intermediate state where only the α -helix is formed [56]. After this intermediate state, the β -hairpin and hydrophobic contacts form. The pathway was determined by combining temperature-dependent UV circular dichroism, Fourier transform infrared spectroscopy, two-dimensional infrared spectroscopy, and temperature-jump transient-IR spectroscopy. This folding mechanism shows good agreement with the dominant folding pathway identified in this study. This agreement strongly indicates that our method can serve as a powerful tool to study the folding mechanism of a protein with atomic details. We note that additional sub-optimal folding pathways were also obtained, where the second lowest OM action pathway suggested a different pathway: the β -hairpin folds first, followed by the concurrent formation of α -helix and hydrophobic contacts (Supplementary Fig. S1).”

Additional technical questions:

Is entropy of paths accounted for in optimization scheme?

The OM action used in this study takes into account the entropy of a trajectory. The second derivative term in Eq. (3) is related to trajectory entropy connected with fluctuations (Ref. 46).

What is importance of time steps and how was it checked?

What is the sensitivity of the method to the weighting parameters of Eq (6)?

For an Action-CSA calculation, there are 10 adjustable parameters, which are listed in a newly added Supplementary Table 1. It should be noted that the parameters for the calculations presented in the current manuscript were not extensively optimized. As a preliminary study, we performed additional alanine dipeptide simulations with various timestep values and μ_E values. Overall, the alanine dipeptide results show almost identical results. For FSD-1, we observe that too big a timestep can lead to unphysical crossing events between peptide chains. In near future, we plan to perform subsequent benchmark simulations to identify the effects of input parameters on pathway search results. However, a detailed discussion on parameter optimization for Action-CSA calculations is out of the scope of the current manuscript, and requires a multi-year effort. Based on this comment, we added the following sentences at the end of the first paragraph on page 14:

“An Action-CSA calculation requires 10 adjustable parameters, and they are listed in Supplementary Table S1. The parameters for the calculations presented in this study were not extensively optimized. Rigorous optimization of the parameters is out of the scope of this study, and requires a series of subsequent benchmark studies.”

=====

Reviewer #3 (Remarks to the Author):

The authors present a global search strategy for finding multiple reaction pathways that are founded on a maximization of the Onsager-Machlup (OM) action. They present a new way to apply the OM formula in its discretized way in combination with CSA (Conformational space annealing). CSA is used to optimize the OM action. Therefore, only the first derivatives with respect to the potential energy are needed. This is an improvement over older implementations, which used second derivatives. (related to formula 3 in the manuscript). Related to this, Fujisaki et al. (J. Chem. Phys. 132, 134101 (2010)) presented some work by using the replica exchange approach for the Bolhuis potential. The authors present an applicable tool for applying it to real chemistry problems. Their bank approach seems to be indeed an intelligent way to guarantee the diversity within the whole procedure. For all their tests, they present a trustworthy proof either computationally through Langevin Dynamic simulations or experiments.

We greatly appreciate for reviewer's positive and encouraging comments. As the reviewer pointed out, the first improvement of our method is the combined use of two actions: using classical action to optimize trial pathways and to rank the optimized pathways using Onsager-Machlup action. By using the equation derived by Miller and Predescu, Eq. (4), Action-CSA finds multiple pathways with low OM action efficiently without second derivatives. Due to the efficiency of Action-CSA, the method can be applied to study various real biological and chemical reactions.

In this way, the paper seems to be an improvement over existing approaches and is recommended for publication.

Related to equation 1 one question remains open. Would it be the better choice to introduce mass weighted coordinates directly? For Brownian dynamics (overdamped Langevin dynamics) the second derivatives can be neglected if the mass goes to zero, then the direct use of m would be replaced and already be included in x .

We agree with the reviewer's comment. Because we are interested in a long-time behavior of a reaction, the second derivatives can be neglected. Also, the use of mass-weighted coordinate will indeed simplify the expression. Thus, we removed the mass term in the related equations in the manuscript and specified that x represents the mass-weighted coordinate in the first paragraph on page 11. We thank the reviewer for this useful suggestion.

REVIEWERS' COMMENTS:

Reviewer #1 (Remarks to the Author):

The authors responded well to my critics and I believe the manuscript is ready for publication.

Reviewer #2 (Remarks to the Author):

Recommendation: Publish after minor revision.

The authors have clarified some important issues and the additional validation of their folding pathway is impressive, but some points still have not been addressed adequately. Using the [numbering/notation] from previous review by reviewer #2:

[(2) Non-action-based path-sampling methods.]

In their response, the authors re-stated *procedural* novelties of their method, which are quite different from novel *capabilities*. As noted in the original review, a host of path-sampling methods have been applied to difficult protein problems. Such methods easily deal with large barriers. The authors are wrongly dismissive of such methods - this demonstrates a lack of understanding or poor standards of scholarship, and should not be acceptable for publication to my view.

[Additional technical question on path entropy]

I think the authors misunderstood the question, which is not regarding entropy generation in a trajectory, but rather the effective entropy associated with a set of trajectories following a given pathway - sometimes called a tube of trajectories. The overall probability of a pathway/tube will be affected by the diversity/degeneracy/size of the tube. I don't believe that is part of the OM action for a single trajectory - after all, the action of a trajectory is analogous to the energy of a configuration in an equilibrium ensemble. Does the current study account for the entropy of trajectory diversity within a tube? In general, path sampling methods should automatically account for this.

Reviewer #3 (Remarks to the Author):

The authors took all our suggestions into account. The comment of referee 2 who questioned the uniqueness of the algorithm is in parts true, however this holds for nearly all algorithms. Considering that the improvements made with respect to previous approaches (e.g. Fujisake et al. JPC 132, 134101 (2010)) are significant the paper should be published.

REVIEWERS' COMMENTS:

Reviewer #1 (Remarks to the Author):

The authors responded well to my critics and I believe the manuscript is ready for publication.

Reviewer #2 (Remarks to the Author):

Recommendation: Publish after minor revision.

The authors have clarified some important issues and the additional validation of their folding pathway is impressive, but some points still have not been addressed adequately. Using the [numbering/notation] from previous review by reviewer #2:

[(2) Non-action-based path-sampling methods.]

*In their response, the authors re-stated *procedural* novelties of their method, which are quite different from novel *capabilities*. As noted in the original review, a host of path-sampling methods have been applied to difficult protein problems. Such methods easily deal with large barriers. The authors are wrongly dismissive of such methods - this demonstrates a lack of understanding or poor standards of scholarship, and should not be acceptable for publication to my view.*

The second reviewer questions about *procedural* novelties vs. novel *capabilities*. We are afraid that he is unreasonably critical about the novelty of our method while the other two reviewers (especially as the third reviewer adequately pointed out in his second report) acknowledge that the improvements made by our method are significant.

However, in order to tone down the novelty of our method, we modified the following phrases in the revised manuscript:

"Currently, there exist no practical methods that can efficiently explore and produce multiple reaction pathways connecting two given end states of a complex system. The objective of this work is to present such a method."

to

"Although several such methods have been suggested [1–8], exploring and producing multiple reaction pathways of a complex system remains a challenge. The objective of this work is to present a method that can efficiently explore and produce multiple reaction pathways connecting two end states."

[Additional technical question on path entropy]

I think the authors misunderstood the question, which is not regarding entropy generation in a trajectory, but rather the effective entropy associated with a set of trajectories following a given pathway - sometimes called a tube of trajectories. The overall probability of a pathway/tube will be affected by the diversity/degeneracy/size of the tube. I don't believe that is part of the OM action for a single trajectory -

after all, the action of a trajectory is analogous to the energy of a configuration in an equilibrium ensemble. Does the current study account for the entropy of trajectory diversity within a tube? In general, path sampling methods should automatically account for this.

About the diversity/degeneracy/size of the tube, the Hessian term in Eq (3) exactly accounts for this. The harmonic expansion around local minima, that the Hessian provides, determines the diversity/degeneracy/size of the tube in an approximate way. In the work by Adib (Adib, A. B. *J. Phys. Chem. B* 112, 5910–5916 (2008), ref #66 in the revised manuscript), it was shown that when a trajectory motif is located at local minima or points of very low gradients in comparison to curvatures, the OM action with the Hessian term corresponds to the trajectory free energy itself. Also, it was shown that, for the purposes of reweighting and sampling diffusive paths, two actions with and without the Hessian term are equivalent. However, for problems concerned with the most likely trajectory motif, it was argued that the term contains the relevant entropic corrections connected with fluctuations about the trajectory motif, and hence should be favored over the action without the Hessian term in direct minimization studies. To clarify this point, we added the following sentences in the Methods section.

In the original formula of action derived by Onsager and Machlup, the last term of Eq. 3 was absent [26, 27]. It was shown that, for the purposes of reweighting and sampling diffusive pathways, two OM actions with and without the Hessian term are equivalent. However, for the purpose of finding the most probable trajectory motif, the term should be considered because it represents the entropic corrections connected with fluctuations and the neighborhood of a given trajectory motif, which is also represented as a tube around the motif [66, 67].

Reviewer #3 (Remarks to the Author):

*The authors took all our suggestions into account. The comment of referee 2 who questioned the uniqueness of the algorithm is in parts true, however this holds for nearly all algorithms. Considering that the improvements made with respect to previous approaches (e.g. Fujisake et al. *JPC* 132, 134101 (2010)) are significant the paper should be published.*